# Lung-Derived Selectins Enhance Metastatic Behavior of Triple Negative Breast Cancer Cells

**DOI:** 10.3390/biomedicines9111580

**Published:** 2021-10-30

**Authors:** Sami U. Khan, Ying Xia, David Goodale, Gabriella Schoettle, Alison L. Allan

**Affiliations:** 1Department of Anatomy & Cell Biology, Western University, London, ON N6A 3K7, Canada; sami.k@live.ca; 2Robarts Research Institute, Western University, London, ON N6A 3K7, Canada; yxia3@uwo.ca; 3London Health Sciences Centre, London Regional Cancer Program, London, ON N6A 5W9, Canada; david.goodale@lhsc.on.ca; 4Schulich School of Medicine & Dentistry, Western University, London, ON N6A 5C1, Canada; gschoettle2023@meds.uwo.ca; 5London Health Sciences Centre, London Regional Cancer Program, Departments of Oncology and Anatomy & Cell Biology, Lawson Health Research Institute, Western University, London, ON N6A 5W9, Canada

**Keywords:** breast cancer, metastasis, lung microenvironment, E-selectin, L-selectin, P-selectin, cd44, bimosiamose

## Abstract

The lung is one of the deadliest sites of breast cancer metastasis, particularly for triple negative breast cancer (TNBC). We have previously shown that the lung produces several soluble factors that may enhance the metastatic behavior of TNBC, including E-, L-, and P-selectin. In this paper, we hypothesize that lung-derived selectins promote TNBC metastatic behavior and may serve as a potential therapeutic target. Lungs were isolated from mice and used to generate lung-conditioned media (CM). Lung-derived selectins were immunodepleted and TNBC migration and proliferation were assessed in response to native or selectin-depleted lung-CM. A 3D ex vivo pulmonary metastasis assay (PuMA) was used to assess the metastatic progression of TNBC in the lungs of wild-type versus triple-selectin (ELP^-/-^) knockout mice. We observed that individual lung-derived selectins enhance in vitro migration (*p* ≤ 0.05), but not the proliferation of TNBC cells, and that ex vivo metastatic progression is reduced in the lungs of ELP^-/-^ mice compared to wild-type mice (*p* ≤ 0.05). Treatment with the pan-selectin inhibitor bimosiamose reduced in vitro lung-specific TNBC migration and proliferation (*p* ≤ 0.05). Taken together, these results suggest that lung-derived selectins may present a potential therapeutic target against TNBC metastasis. Future studies are aimed at elucidating the pro-metastatic mechanisms of lung-derived selectins and developing a lung-directed therapeutic approach.

## 1. Introduction

Breast cancer is the most diagnosed cancer and a leading cause of death in women worldwide [1]. More than 90% of these deaths are due to metastasis, as current treatments are non-curative in the metastatic setting [2,3,4]. The lung is one of the most common and deadly sites of breast cancer metastasis, particularly in patients with aggressive triple negative (TN) disease [5,6,7]. Lung metastasis often occurs within 5 years of the initial diagnosis and leads to significant morbidity and mortality [5,7,8,9,10]. Patients who present with either solitary or small numbers of lung metastases (termed “oligometastatic” disease [11]) have demonstrated a survival benefit from targeted resection and/or stereotactic ablative radiation of these lesions [12,13]. However, more advanced lung metastases remain difficult to treat, with an estimated 60–70% of patients who die of breast cancer having lung metastasis [14]. This important clinical problem highlights the need to characterize the fundamental cellular and molecular drivers of this process in order to develop new therapeutic strategies.

We have previously identified several soluble factors produced by the normal lung that promote TN breast cancer metastatic behavior [15]. This work has been facilitated by the development of a novel 2D ex vivo model system for investigating the influence of organ-derived soluble factors on breast cancer metastatic behavior, whereby organs representing common sites of breast cancer metastasis (i.e., lung) are isolated from healthy mice and used to generate organ-conditioned media (CM) [16]. In particular, we have observed that aggressive TN breast cancer cells expressing high levels of the CD44 receptor (standard form; CD44S) have a propensity to migrate towards the lung-CM in vitro [15] and metastasize to lung in vivo [17]. Previous protein array analysis of lung-CM also revealed the lung as an important source of soluble proteins known to interact with CD44, including E-, L-, and P-selectin [15,18,19,20,21].

Selectins belong to a three-member family of type I cell-surface glycoproteins involved in mediating cell–cell adhesion and migration [22]. There is high sequence homology between E-, L-, and P-selectin in all domains except the transmembrane and cytoplasmic domains. E-selectin is expressed by endothelial cells [22,23]; L-selectin is expressed by leukocytes [24]; and P-selectin is expressed by platelets and endothelial cells [25]. All three selectins have been shown to interact with CD44 in order to facilitate both normal leukocyte adhesion and rolling as well as cell–cell adhesion in pathological conditions, such as cancer metastasis [21,26,27,28,29]. Selectins exist not only as membrane-bound proteins, but also as cleaved soluble proteins. Cleavage is believed to be proximal to the membrane spanning domain [30,31,32], and in vitro studies suggest that soluble E-, L-, and P-selectin remain functionally active after being shed from the cell surface [33,34,35].

Experimental mouse models deficient in one or more selectins have been developed [36,37,38,39]. Phenotypically, individual selectin deficiencies in either E-, L-, or P-selectin do not cause infertility or embryonic lethality. However, double (EP^-/-^ or PL^-/-^) or triple (ELP^-/-^) knockout mice experience health issues, such as mucocutaneous infections, due to alterations in leukocyte homeostasis and recruitment [37]. While these deficiencies may negatively affect immune cell recruitment, evidence suggests that they may also be advantageous to disruption of metastasis. Using an experimental model of breast cancer, it has been observed that lung metastasis is reduced in E^-/-^ and EP^-/-^ mice compared to wild-type mice [40]. Similarly, in a colon cancer model, P^-/-^, L^-/-^, or PL^-/-^ mice demonstrate reduced lung metastasis compared to the wild-type control, with the double selectin knockouts having the lowest metastatic burden [41].

In the clinical setting, it has been reported that serum levels of E-selectin are significantly higher in patients with advanced breast cancer that have distant metastasis compared to earlier stage patients [42,43,44], and that high E-selectin serum concentrations are prognostic of a worse outcome [44,45]. While previous studies have not established a role for L-selectin in breast cancer, acute myeloid leukemia patients with elevated plasma L-selectin levels have poorer outcomes [46], and increased concentrations of L-selectin have also been observed in the serum of patients with metastatic versus nonmetastatic bladder cancer [47]. Soluble P-selectin has been detected in increased amounts in the sera of breast cancer and colon cancer relative to healthy controls [48,49].

While these studies provide experimental and clinical evidence supporting the importance of selectins in cancer progression, the functional role of soluble selectins in breast cancer metastatic behavior remains poorly understood. In the current study, we test the hypothesis that lung-derived selectins promote TN breast cancer metastatic behavior and may serve as a potential therapeutic target. Using lung-conditioned media and a 3D ex vivo pulmonary metastasis assay (PuMA), we demonstrate that individual lung-derived selectins enhance in vitro migration, but not proliferation of TN breast cancer cells. In addition, we demonstrate that ex vivo metastatic progression is reduced in the lungs of ELP^-/-^ mice compared to wild-type mice and that treatment with the pan-selectin inhibitor bimosiamose reduces lung-specific TN breast cancer migration and proliferation in vitro. Taken together, these results suggest that lung-derived selectins may present a potential therapeutic target against TN breast cancer metastasis.

## 2. Materials and Methods

### 2.1. Cell Culture, Reagents, and Model Selection

Triple negative human breast cancer cells (MDA-MB-231, SUM149, and SUM159) were cultured and maintained at 37 °C and 5% CO_2_. MDA-MB-231 human breast cancer cells (ATCC, Manassas, VA, USA) were maintained in a Dulbecco’s Modified Eagle Medium/Nutrient Mixture F-12 (DMEM/F12) media + 10% fetal bovine serum and passaged using 1 mM ethylene diamine tetraacetic acid (EDTA). SUM149 and SUM159 human breast cancer cells (Asterand Inc., Detroit, MI, USA) were maintained in HAMS F-12 media supplemented with 5% FBS, 5 μg/mL insulin, 1 μg/mL hydrocortisone, and 10 mM HEPES and passaged using 0.25% trypsin/EDTA in citrate saline. Additional cell lines were also engineered to express tdTomato (MDA-MB-231) or mCherry (SUM159) via lentiviral transduction. All media/supplements were from Invitrogen (Carlsbad, CA, USA) and FBS was from Sigma-Aldrich (St Louis, MO, USA). Cell lines were authenticated via third party testing (IDEXX BioAnalytics, Columbia, MO, USA). The pan-selectin inhibitor bimosiamose (TBC1269) [50] was obtained from MedKoo Biosciences (Morrisville, NC, USA).

All the TNBC cell lines that were used for this study (MDA-MB-231, SUM159, and SUM149) have metastatic potential in vivo; the SUM149 cells are weakly metastatic whereas the MDA-MB-231 and SUM159 cells are highly metastatic. We wanted to test the effect of selectins on breast cancer cells that had a range of metastatic potential. We therefore chose to use the SUM149 and MDA-MB-231 cells for the in vitro experiments to provide this spectrum. While SUM149 cells can be used in the ex vivo PuMA (described below), they require a much longer time window for metastatic progression and so are difficult to compare directly with the MDA-MB-231 model. We therefore chose to use SUM159 cells as a second TNBC model for the ex vivo studies.

### 2.2. Lung-Conditioned Media

Lung-conditioned media were generated as described previously [15]. Briefly, healthy 5–7 week old female athymic nude mice were purchased and maintained in accordance with the Canadian Council of Animal Care under a protocol approved by the University of Western Ontario Animal Care Committee (#2016-091). Mice were euthanized by CO_2_ inhalation; lungs were harvested, washed 3 times in ice-cold PBS, and minced into ~ 1 mm^3^ fragments. Lung fragments were weight-normalized by re-suspension in a 4:1 media to tissue (v/w) ratio in DMEM:F12 supplemented with Mito+ serum extender (1X; BD Biosciences, Mississauga, ON, Canada; basal media) + 50 U/mL penicillin/50 μg/mL streptomycin (Invitrogen) and incubated at 37 °C, 5% CO_2_ for 24 h. The resulting lung-conditioned media (CM) were further diluted 4-fold in basal media, separated from tissue fragments by centrifugation at 900× *g* for 15 min at 4 °C and filtered through 0.22 μm filters (Corning Inc., Corning, NY, USA). The resulting lung-CM were used immediately for the experiments described below, or aliquoted and stored at −80 °C for later use (Appendix A).

### 2.3. Selectin Enzyme Linked Immunosorbent Assay (ELISA)

To assess the presence and concentration of selectins in the lung-CM, ELISA kits for soluble mouse E-selectin/CD62E, soluble mouse L-selectin/CD62L, and soluble mouse P-selectin/CD62P (R&D Systems, Minneapolis, MN, USA) were used. Basal media, native lung-CM, and lung-CM depleted of E-, L-, or P-selectin (described below and Appendix A) were added to pre-coated microplates and ELISAs were carried out as per the manufacturer’s instructions to measure the concentration of E-, L-, and P-selectin in the lung-CM and efficiency of immunodepletion. A four-parameter logistic (4-PL) standard curve was generated for each experiment using GraphPad Prism 6.0 (GraphPad Software, San Diego, CA, USA) and used to determine the exact protein concentration in each sample.

### 2.4. Selectin Immunodepletion

E-, L-, and P-selectin were individually immunodepleted from the lung-CM using Dynabeads^®^ Protein G (Life Technologies, Carlsbad, CA, USA) and antibodies against specific soluble selectins as detailed in Appendix A. Dynabeads^®^ were suspended in 200 μL of wash buffer (PBS + 0.02% Tween 20) to be equilibrated and, then, placed on the magnetic rack for 2 min for their separation from the solution. The wash buffer was carefully removed, replaced with wash buffer + appropriate antibody, mixed by pipetting, and allowed to nutate for 20 min at RT. Following nutation, the tubes were placed on the magnetic rack for 2 min to separate the antibody-bound beads from the solution. The supernatant was carefully removed and the antibody-bound beads were washed 3× before incubation with the lung-CM on the nutator for 30 min at RT. Post-nutation, the samples were placed on the magnetic rack for 2 min to separate the selectin/antibody/bead complexes from the now depleted lung-CM. A small aliquot of depleted lung-CM was used for the quantification of protein levels by ELISA and the balance was stored at −80 °C until used as outlined in Appendix A.

### 2.5. Co-Immunoprecipitation and Immunoblotting

Confluent 100 mm dishes of MDA-MB-231 human breast cancer cells were washed 3 times with PBS and lysed with 500 μL of lysis buffer (1% NP-40, 25mM Tris-HCl pH 7.5, 100 mM NaCl, 5% Glycerol, and 1x Halt™ Protease Inhibitor Cocktail (Thermo Scientific, Waltham, MA, USA). Cell lysates were collected and incubated on ice for 1 h prior to centrifugation for 10 min at 13,000× *g* and 4 °C. During this period, 20 μL (0.6 μg) of Dynabeads^®^ were equilibrated in 200 μL of wash buffer (25mM Tris-HCl pH 7.5, 100 mM NaCl, and 5% glycerol) and placed on a magnetic rack for 2 min to separate from the solution. The wash buffer was carefully removed and replaced with 500 μL of the MDA-MB-231 cell lysate supernatant. The remaining pellet from the cell lysates was discarded. Two μg each of recombinant mouse E-, L-, or P-selectin Fc chimera protein (R&D Systems, Minneapolis, MN, USA) were added to each tube containing cell lysate/beads or beads only (control), and incubated on a rotator overnight at 4 °C. The samples were then placed on a magnetic rack to separate the beads/bound proteins from the solution. The supernatant was carefully removed, and the remaining beads were resuspended in the wash buffer, washed, and magnetically separated. The wash and separation steps were repeated twice and the beads were suspended in 30 μL of Laemmli buffer and boiled for 5 min. The samples were placed on the magnetic rack for 2 min and the supernatant was collected. Thirty μL of each immunoprecipitated sample was subjected to separation by sodium dodecyl sulfate polyacrylamide gel electrophoresis (SDS-PAGE) and transferred onto polyvinylidene diflouride membranes (PVDF; Millipore, Billerica, MA, USA). The membranes were blocked using 5% skim milk in tris-buffered saline + 0.1% Tween 20 (TBS-T; Sigma, Burlington, MA, USA). Primary and secondary antibody concentrations are listed in Appendix A. Protein expression was visualized using Clarity Max™ Western ECL Substrate (Bio-Rad Laboratories, Hercules, CA, USA).

### 2.6. In Vitro Cell Migration Assays

Breast cancer cell migration was assessed using transwell migration assays. Fluoroblok™ transwell inserts (24-well, 8 μm pore size; Corning Inc., Corning, NY, USA) were coated with 100 μL of gelatin (6 μg/well; Bioshop, Burlington, ON Canada). Human TN breast cancer cells (MDA-MB-231, SUM149, and/or SUM159) were plated on top of the transwells at 5 × 10^4^ cells/well. For lung-derived selectin depletion studies, the lower compartment of the transwell chambers contained either basal media (negative control), native lung-CM, or lung-CM depleted of E-, L-, or P-selectin. For rescue conditions, an amount of recombinant E-,L, or P-selectin protein (R&D Systems, Minneapolis, MN, USA) equal to the calculated average protein concentration of each immunodepleted selectin was added back to the depleted lung-CM (E-selectin (0.25 ng/mL) and L-selectin (7.05 ng/mL)) and P-selectin (10.39 ng/mL). For selectin inhibitor studies, the lower compartment contained basal media, basal media + 10% FBS, or native lung-CM containing 1 mM of bimosiamose or dimethyl sulfoxide (DMSO; vehicle control). The plates were incubated for 18 h at 37 °C and 5% CO_2_ to allow migration. After 18 h, the transwell chambers were fixed with 1% glutaraldehyde in PBS for 20 min, washed, and the non-migrated cells were removed from the inner portion of the membrane using a cotton swab. The membranes were carefully cut out, placed on microscope slides, and mounted using ProLong^®^ Gold Antifade Mountant with DAPI (Life Technologies, Carlsbad, CA, USA). Five high powered fields of view (HP-FOV) were captured for each membrane, and the mean number of migrated cells/FOV was calculated using ImageJ software (NIH, Bethesda, MD, USA).

### 2.7. In Vitro Cell Proliferation Assays

Breast cancer cell proliferation was assessed using bromodeoxyuridine (BrdU) incorporation assays. Human TN breast cancer cells (MDA-MB-231, SUM149, and/or SUM159; 1 × 10^4^ cells/well) were plated on 8-well Lab-tek™ chamber slides (Thermo Scientific, Waltham, MA, USA), incubated for 24 h to allow cell adhesion, washed 1× with PBS, and serum starved for 72 h. For lung-derived selectin depletion studies, media were changed to either basal media, positive control media (basal media + 10% FBS), native lung-CM, or lung-CM depleted of E-, L-, or P-selectin. For selectin inhibitor studies, media were changed to either basal media, basal media + 10% FBS, or native lung-CM containing 1 mM of bimosiamose or DMSO. After 24 h, the cells were exposed to BrdU (5 μg/mL; Amersham Cell Proliferation Labelling Reagent; GE Healthcare, Piscataway, NJ, USA) for 30 min to allow incorporation into newly synthesized DNA. The cells were washed with PBS, fixed with 10% neutral-buffered formalin (Fisher Scientific), permeabilized using 0.1% Triton X-100 (Sigma), and treated with 2N HCl to denature DNA. To minimize non-specific binding, the slides were blocked for 30 min in 5% BSA + 0.1% Triton X-100 in PBS. For detection, the anti-BrdU primary antibody (BD Biosciences), diluted at 1:75 in 5% BSA/0.1% Triton X-100 in PBS, was added and the slides were incubated overnight at 4 °C. A FITC-conjugated anti-mouse IgG secondary antibody (1:100 dilution; Vector Laboratories, Burlington, ON, Canada) was added for 1 h at RT. The slides were washed and mounted with ProLong Gold Antifade Mountant with DAPI (Invitrogen, Waltham, MA, USA), and allowed to cure overnight in the dark at RT prior to the analysis of five HP-FOV. The percentage of BrdU-positive cells to total nuclei were calculated using Image J.

### 2.8. Ex Vivo Pulmonary Metastasis Assay (PuMA)

An ex vivo pulmonary metastasis assay (PuMA) was used for real-time assessment of metastatic progression in intact lung tissue as described previously [51,52,53] and in Appendix A. Wild-type or triple-selectin knockout (ELP^-/-^) mice (B6;129S-Sele^tm1Hyn^ Sell^tm1Hyn^ Selp^tm1Hyn^/J; JAX stock #003807; The Jackson Laboratory, Bar Harbour, ME, USA) [39] were purchased and bred/maintained in accordance with the Canadian Council of Animal Care, under a protocol approved by the University of Western Ontario Animal Care Committee (#2016-091). Red fluorescent human breast cancer cells (MDA-MB-231-TdTomato or SUM159-mCherry) were harvested, suspended in Hank’s balanced salt solution (HBSS; Invitrogen, Waltham, MA, USA), and delivered by tail vein injection to 5–7 week old female wild-type or ELP^-/-^ mice (*n* = 4/group; 1 × 10^5^ cells/mouse). Within 15 min of the cell injection, the mice were euthanized by CO_2_ inhalation. Using sterile surgical conditions, the trachea was snipped and cannulated with an 18G blunt needle. The lungs were infused under gravitational pressure with 1.2 mL of equal amounts of Lung Media 1 (Appendix A) + low melting agarose solution (0.6%, 40 °C). The trachea, lungs, and heart were carefully removed en bloc and immediately placed in ice-cold PBS containing 100 U/mL penicillin and 100 μg/mL streptomycin, and incubated at 4 °C for 20 min to solidify. Transverse lung sections (1–2 mm thick; 12–20 sections/lung) were generated from each lobe using a scalpel blade and sections were placed on a sterile piece of Gelfoam (~1 × 1 cm) that had been preincubated for 1–2 h in a 6-well plate with Lung Media 2 (Appendix A). The lung sections were grown in culture for 21 days at 37 °C in 5% CO_2_. The media were replaced every other day and lung tissue sections turned over carefully with tweezers. At 0, 7, 14, or 21 days post-injection, seeded lung sections were removed from the culture and fixed overnight in 10% buffered formalin phosphate (Fisher Scientific, Waltham, MA, USA) + 25% sucrose (weight/volume) to preserve fluorescence. The sections were rinsed 3× with PBS, placed on a glass slide, and gently cover-slipped. Images were acquired using an upright Nikon A1R confocal microscope at 20X objective (Nikon), with a 591 nm emission laser (Melles Griot, Carlsbad, CA, USA). Three separate lung sections were imaged per timepoint with five FOV taken per lung section. Metastatic progression of breast cancer cell populations within the lung was determined by measuring the mean fluorescent area per FOV for each section of lung (μm^2^) using Image J. Data were normalized to day 0 to control for variability in cellular delivery during tail vein injection.

### 2.9. Data Analysis

A minimum of three biological replicates were performed for each experiment, with at least three technical replicates within each experiment for transwell migration and BrdU incorporation assays. All statistical analyses were performed using GraphPad Prism 6.0 (San Diego, CA, USA), and data are presented as mean ± standard error of the mean (SEM). A one-way analysis of variance (ANOVA) was used to compare means between experimental groups, followed by a Tukey’s post-hoc test. In all cases, values of *p* ≤ 0.05 were considered statistically significant.

## 3. Results

### 3.1. E-, L-, and P-Selectin Are Present in Lung-CM and Can Interact with CD44 Expressed by Triple Negative Breast Cancer Cells

Our previous studies have demonstrated that conditioned media generated from the lungs of healthy mice can be used as a model of the soluble lung microenvironment and its role in influencing breast cancer metastatic behavior [15]. Using a protein array analysis, we identified more than 70 lung-derived proteins in lung-CM that are not present in basal media [15]. In particular, E-, L-, and P selectin were identified as proteins for further investigation based on their previous association with cancer metastasis [18,19,20,21]. Using ELISA, we observed that E-selectin (SELE; Figure 1a), L-selectin (SELL; Figure 1b), and P-selectin (SELP; Figure 1c) are all present in lung-CM at significantly higher concentrations than in basal media (*p* < 0.05). Individual selectins could be successfully immunodepleted, resulting in lung-CM that had significantly reduced levels of E-, L-, or P-selectin compared to non-depleted lung-CM (*p* < 0.05). In the absence of a specific antibody, Dynabeads^®^ Protein G alone had no significant effect on the immunodepletion of selectins (Figure 1a–c).

Soluble selectins have been demonstrated to interact with CD44 in order to exert their functional effects on target cells [21,26,27,28,29]. We have previously observed that TN breast cancer cells (including the MDA-MB-231, SUM149, and SUM159 models used in this study) express high levels of the standard form of CD44 (CD44S) on their cell surface [15], suggesting a potential mechanism by which lung-derived selectins might influence breast cancer cell behavior. To confirm whether murine selectins could interact with the CD44S expressed by human breast cancer cells in vitro, soluble forms of mouse E-, L-, and P-selectin IgG1 chimeric proteins were used for immunoprecipitation experiments. A single band was observed in each lane containing the recombinant selectin protein that immunoprecipitated with the Dynabead^®^ Protein G (Figure 1d), indicating that the recombinant chimeric selectins could interact directly with the Dynabead^®^ Protein G without additional antibodies. In co-immunoprecipitation experiments, we observed that bands corresponding to proteins of approximately 90 kDa in weight reacted with the CD44S antibody and were observed in each of the E-, L-, and P-selectin immunoprecipitates (Figure 1e). The protein size detected was consistent with the band observed from the MDA-MB-231 cell lysate input control. This indicates that murine E-, L-, and P-selectin can interact with the CD44S expressed by human TN breast cancer cells.

### 3.2. Lung-Derived Selectins Enhance Triple Negative Breast Cancer Cell Migration

To determine the functional influence of individual lung-derived selectins on TN breast cancer migration, a transwell migration assay was used. MDA-MB-231 (Figure 2a–c) or SUM149 (Figure 2d–f) human TN breast cancer cells were exposed to basal media (negative control), native lung-CM, or lung-CM depleted of E-, L-, or P-selectin. MDA-MB-231 and SUM149 human breast cancer cells demonstrated a significantly increased migration towards native lung-CM relative to basal media control (*p* < 0.05). This increased migration was abrogated in the presence of lung-CM depleted of either E-selectin (Figure 2a,d), L-selectin (Figure 2b,e), or P-selectin (Figure 2c,f) (*p* < 0.05). Selectin-dependent migration could be rescued by the addition of individual recombinant soluble mouse selectins at a level that restored respective selectin protein levels to those detected in the native lung-CM (Figure 2a–f). Taken together, these results indicate that lung-derived E-, L-, and P-selectin enhance migration of TN breast cancer cells.

### 3.3. Lung-Derived E-,L-, or P-Selectin Do Not Individually Influence Breast Cancer Cell Proliferation

To determine whether individual lung-derived selectins also have a functional role in cell proliferation, BrdU incorporation assays were used. MDA-MB-231 (Figure 3a–c) and SUM149 (Figure 3d–f) TN human breast cancer cells were exposed to basal media, native lung-CM, or lung-CM depleted of E-, L-, or P-selectin, and BrdU incorporation was assessed. Consistent with our previous studies [15], we observed an increase in TN breast cancer cell proliferation in response to native lung-CM versus basal media control (*p* < 0.05) (Figure 2a–f). However, the proportion of BrdU-positive cells was unaffected by the individual depletion of E-selectin (Figure 3a,d), L-selectin (Figure 3b,e), or P-selectin (Figure 3c,f) from the lung-CM, suggesting that lung-derived E-,L-, or P-selectin do not individually influence breast cancer cell proliferation (Figure 2a–f).

### 3.4. Loss of E-, L-, and P-Selectin in the Lung Reduces Triple Negative Breast Cancer Progression in the Ex Vivo Pulmonary Metastasis Assay (PuMA)

We next wanted to assess the importance of lung-derived selectins in a more physiologically relevant model of breast cancer metastatic progression using the ex vivo pulmonary metastasis assay (PuMA; Appendix A). Because our initial in vitro experiments demonstrated that depletion of each individual selectin effectively reduced breast cancer migration in a similar manner, we were interested in determining if a concomitant deficiency in all three selectins would lead to reduced metastatic progression in the PuMA in the context of triple-selectin (ELP^-/-^) knockout mice [39]. We have previously observed that SUM149 cells are only weakly metastatic both in vivo and in the ex vivo PuMA (unpublished data), so for these experiments we utilized the more metastatic SUM159 cell line instead of SUM149. MDA-MB-231 and SUM159 TN breast cancer cells were seeded into the lungs of wild-type or ELP^-/-^ mice. We observed that ex vivo breast cancer metastatic progression is reduced in the lungs of ELP^-/-^ mice relative to the wild-type control after 21 days (*p* < 0.05) (Figure 4), suggesting that selectins are important for lung metastasis of TN breast cancer and may serve as a potential therapeutic target.

### 3.5. The Pan-Selectin Inhibitor Bimosiamose Reduces In Vitro Lung-Specific TN Breast Cancer Migration and Proliferation

Finally, to begin exploring this therapeutic potential, we used bimosiamose (TBC-1269), a small-molecule non-oligosaccharide inhibitor of E-, L-, and P-selectin binding [54]. MDA-MB-231 (Figure 5a,b) and SUM159 (Figure 5c,d) TN human breast cancer cells were exposed to basal media, regular growth media (10% FBS; positive control), or native lung-CM, and transwell migration and BrdU incorporation was assessed in response to DMSO (dimethyl sulfoxide; vehicle control) or 1 mM bimosiamose. The growth media contain 10% FBS, a potent mitogenic and chemoattractant containing numerous growth factors and chemokines. It was used as a non-specific positive control in this experiment and, as expected, overall TNBC cell proliferation and migration was higher than that observed in response to lung-CM (Figure 5). Consistent with our observations in the ex vivo PuMA model (Figure 4), we observed that pan-selectin inhibition results in reduced in vitro TNBC metastatic behavior, including a decrease in both migration (Figure 5a,c) and proliferation in the presence of lung-CM (Figure 5b,d) (*p* < 0.05) down to the level of the basal media control. Interestingly, this inhibition only occurred in the presence of lung-CM, but not normal growth media, suggesting that the potential therapeutic effect of bimosiamose in reducing TNBC metastatic behavior may be lung-specific.

## 4. Discussion

Despite significant advances in early detection and treatment, breast cancer remains the most commonly diagnosed cancer and a leading cause of cancer-related deaths among women worldwide [1]. The majority of these deaths are due to metastasis, which remains incurable. In particular, patients with the aggressive triple negative (TN) disease demonstrate an increased propensity for lung metastasis, leading to poor outcomes [5,6,7,8,9,10]. Paget’s “seed and soil” hypothesis of metastasis was originally proposed in 1889 and postulates that the formation of metastatic tumors at distant sites, such as the lung, is related not only to the intrinsic properties of the cancer cell (the “seed”), but also to factors found in the organ microenvironment (the “soil”) that allow the tumor cell to survive and grow [55]. Although the presence and expression of selectins in the lung “soil” has been well-established [15,56,57], their specific role in TN breast cancer metastatic behavior is poorly understood. In the current study, we demonstrate that individual lung-derived selectins are necessary (but not sufficient) to promote in vitro migration, but do not individually have an effect on proliferation of TN breast cancer cells. In addition, ex vivo metastatic progression is reduced in the lungs of ELP^-/-^ mice compared to wild-type, and treatment with a pan-selectin inhibitor reduces lung-specific TN breast cancer migration and proliferation in vitro. To the best of our knowledge, this study is the first in the literature to examine the interaction with and the response of breast cancer cells to soluble selectins in the context of the lung microenvironment. Our novel findings suggest that lung-derived selectins may present a potential therapeutic target against TN breast cancer metastasis. 

We observed that individual lung-derived E-, L-, or P-selectin play a functional role in the migration of TN breast cancer cells. It is notable that soluble E-selectin depletion produces a similar reduction in migration to that seen with the depletion of soluble L-selectin and P-selectin, despite being present at an approximately 50-fold lower concentration. This could potentially be explained by the fact that, although E- and P-selectin have both been previously linked to the lung metastasis of different cancer types [40,57,58,59], evidence supports a particular role for the interaction of soluble E-selectin with breast cancer cells to promote metastatic behavior [56,58,60], particularly in the context of CD44+ breast cancer cells [58,60].

With regard to the lack of influence of individual lung-derived selectins on TNBC proliferation, our previously published study [15] identified 67 other soluble proteins that were present in lung-CM compared to basal media. These proteins include several growth factors, any of which could be contributing to the observed proliferation of TNBC cells in response to the lung-CM that is not reduced upon depletion of individual selectins. Specifically in the context of CD44 ligands, we observed that the lung-conditioned media contains osteopontin (OPN) and basic fibroblast growth factor (bFGF), both of which influence TNBC cell proliferation [15]. However, although each individual selectin does not appear to influence proliferation on its own, pan-inhibition of all three selectins either through genetic knockout or through chemical inhibition with bimosiamose strongly suggests that selectins may collectively influence TNBC proliferation and metastatic progression. Elucidation of the contribution of each selectin and the mechanisms of their interaction with each other and/or other soluble factors produced by the lung during these processes will require additional investigation in the future.

The results of this study and previous investigations by our group and others suggest that lung-derived selectins may be exerting their metastatic influence via interactions between selectins and the CD44 receptor on breast cancer cells [15,26,27,56,60,61,62]. The interaction of E-selectin with CD44 expressed by LS174T colon carcinoma cells was originally hypothesized based on CD44 immunoreactivity with the HECA-452 antibody, which detects sialofucosylated oligosaccharides (a group to which E-selectin binds) [20]. For P- and L-selectin, studies demonstrated an interaction with CD44 by blot rolling assays [18]. However, detection of a direct interaction by co-immunoprecipitation of selectins with CD44 expressed by breast cancer cells has only been reported for E-selectin [60]. We therefore investigated this further, using recombinant mouse soluble selectin-human IgG chimera proteins, which are capable of directly interacting with Dynabead^®^ Protein G. This allowed us to avoid potential issues of immunoprecipitating CD44 or selectins using antibodies that target the ectodomains through which they may interact, since the IgG region that interacts with the Dynabead^®^ Protein G is located near the C-terminus of the chimera, far from the ectodomains. We tested this using soluble mouse selectins, as these were the forms that we hypothesized were found in our lung-CM model. The interaction of CD44 with the mouse selectins that we observe is likely facilitated by the high degree of homology (~72%) shared between human and mouse selectins [22]. It should be noted that depleting the cell lysate of CD44 would have provided additional support for the selectin/CD44 interaction (i.e., the disappearance of the 90 kDa band). Finally, CD44 is also expressed by immune cells, such as T lymphocytes and monocytes, so there is the potential that lung-derived selectins may also play a role in immune infiltration during lung metastasis [63,64].

The observed inhibition of breast cancer metastatic behavior in triple-selectin knockout mice or following pan-selectin chemical inhibition in our ex vivo models of the lung microenvironment provides promising preclinical data supporting the therapeutic value of targeting selectins for the treatment of breast cancer lung metastasis. Future follow-up in vivo studies could test the use of an inhalable system for delivering bimosiamose directly to the lung to treat or prevent metastasis. Drugs (ranging from chemotherapeutics to antibiotics) are frequently adapted as inhalable nanoparticles to facilitate the inhalable delivery of these drugs directly into the lung [65,66]. Inhalable delivery of drugs has many advantages, including high bioavailability, rapid onset of action, and direct delivery to the lungs [67,68]. This approach has been used for a variety of respiratory diseases, including cystic fibrosis, asthma, and tuberculosis in both preclinical mouse models and patients [69,70,71,72]. Additionally, a variety of inhalable drug delivery methods have been adapted to cancer treatment, particularly lung cancers (summarized in [72]). Notably, bimosiamose has previously been adapted to an inhalable delivery method for the treatment of asthma patients [73].

In summary, breast cancer lung metastasis is a major contributor to disease-related mortality, particularly in patients with TN disease. The novel findings presented in this paper support the idea that the lung microenvironment plays an important a role in TN breast cancer metastasis, specifically via lung-derived E-, L-, and P-selectin. In accordance with the “seed and soil” theory, targeting the lung microenvironment and interfering with factors, such as selectins, could be a potential therapeutic strategy. Future studies are therefore aimed at elucidating the pro-metastatic mechanisms of lung-derived selectins and developing a lung-directed therapeutic approach for TN breast cancer via an inhalable drug delivery model. We believe that targeting the constant composition of the lung may be more effective than trying to target the highly heterogeneous population of cells that make up TN breast cancer both within and between patients. In the future, this strategy could contribute to improved treatment of breast cancer in the metastatic setting and help to reduce the burden of this disease.

## Figures and Tables

**Figure 1 biomedicines-09-01580-f001:**
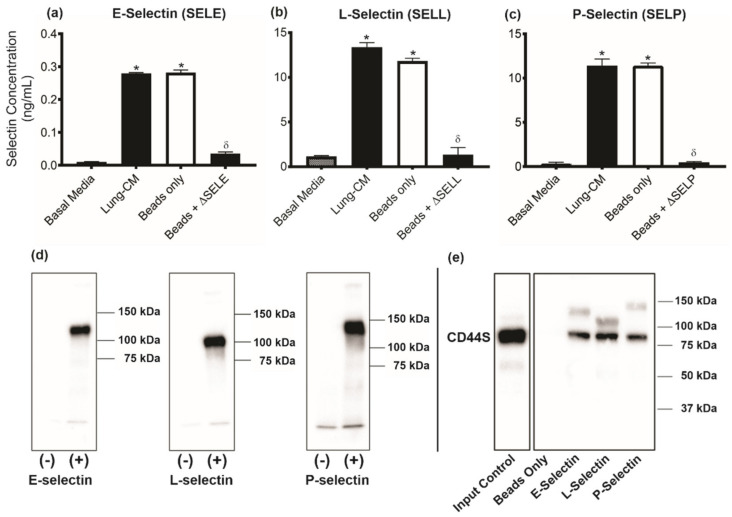
E-, L-, and P-selectin are present in lung-conditioned media (CM) and can interact with CD44 expressed by triple negative breast cancer cells. Lung-CM generated from the lungs of healthy female nude mice were assessed for the presence and concentration of (**a**) E-selectin (SELE), (**b**) L-selectin (SELL), and (**c**) P-selectin (SELP) proteins by ELISA of lung-CM. Immunodepletion (denoted by Δ) was carried out using selectin-specific antibodies coupled with Dynabeads^®^ Protein G. Basal media and lung-CM exposed to beads only were used as controls. Data are presented as mean ± SEM (*n* = 3). * = significantly different than basal media. δ = significantly different than respective non-depleted media. (**d**) Dynabead^®^ Protein G were incubated alone or with murine E-, L-, or P-selectin IgG1 chimeric protein (R&D Systems, Minneapolis, MN, USA). Bound proteins were isolated and analyzed by immunoblotting. (**e**) MDA-MB-231 cell lysates were incubated with Dynabeads^®^ Protein G alone or with murine E-, L-, or P-selectin-human IgG1 chimeric protein. Immunoprecipitates were isolated and immunoblotted. Membranes were visualized by chemiluminescence; representative immunoblots are shown (*n* = 3).

**Figure 2 biomedicines-09-01580-f002:**
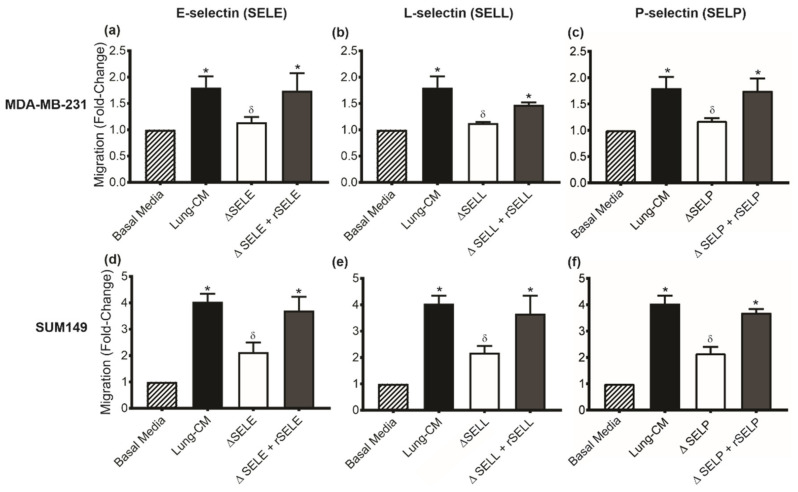
Lung-derived selectins enhance triple negative breast cancer cell migration. (**a**–**c**) MDA-MB-231 and (**d**–**f**) SUM149 human breast cancer cells (5 × 10^4^ per well) were plated on top of gelatin-coated Corning^®^ Fluroblok™ transwells (8-µm pore size) before exposure to one of the following conditions: basal media; native lung-conditioned media (CM); lung-CM depleted of E-selectin, L-selectin, or P-selectin (denoted by ΔSELE, ΔSELL, ΔSELP, respectively); or depleted lung-CM + recombinant (r) SELE, SELL, or SELP (added at the concentration needed to restore protein levels to that observed in native lung-CM). Cells were allowed to migrate for 18 h at 37 °C and 5% CO_2_. Transwells were fixed and stained with DAPI; five high-powered fields of view (HP-FOV) were captured per transwell. Data are presented as mean ± SEM (standard error of the mean); fold-change in migration relative to basal media conditions (*n* = 3). * = significantly different than basal media. δ = significantly different than non-depleted lung-CM.

**Figure 3 biomedicines-09-01580-f003:**
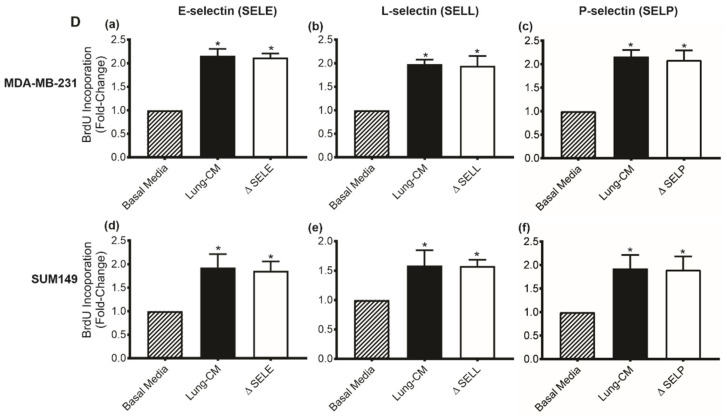
Lung-derived E-,L-, or P-selectin do not individually influence breast cancer cell proliferation. (**a**–**c**) MDA-MB-231 and (**d**–**f**) SUM149 human breast cancer cells were plated at a density of 1 × 10^4^ cells/well on 8-well chamber slides and serum starved for 72 h. Cells were then exposed to basal media, lung-CM, and lung-CM depleted of E-selectin, P-selectin, or L-selectin (denoted by Δ) for an additional 24 h before assessing proliferation based on BrdU incorporation. Five high-powered fields of view (HP-FOV) per well were captured and used to enumerate the percentage of BrdU-positive cells using Image J. Data are presented as mean ± SEM; fold-change in BrdU relative to basal media (*n* = 3). * = significantly different than basal media.

**Figure 4 biomedicines-09-01580-f004:**
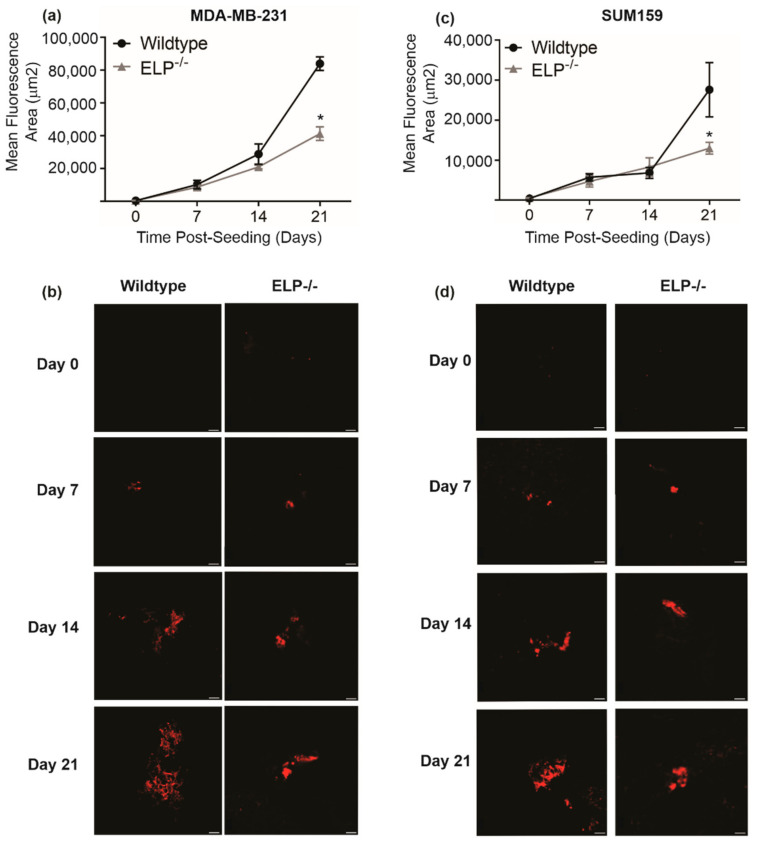
Loss of E-, L-, and P-selectin in the lung reduces triple negative breast cancer progression in the ex vivo pulmonary metastasis assay (PuMA). (**a**,**b**) MDA-MB-231-TdTomato or (**c**,**d**) SUM159-mCherry human TN breast cancer cells were injected into female wild-type or ELP^-/-^ mice (*n* = 4/group/timepoint; 1 × 10^5^ cells/mouse). Lungs were harvested after 15 min and subjected to the PuMA for 21 days. (**a**,**c**) Data are presented as mean ± SEM; fluorescent area (μm^2^) normalized to time 0. * = significantly different than wild-type mice. (**b**,**d**) Representative images are shown for metastatic progression in the (**b**) MDA-MB-231 and (**d**) SUM159 models in wild-type and ELP^-/-^ mice. Scale bars = 100 µm.

**Figure 5 biomedicines-09-01580-f005:**
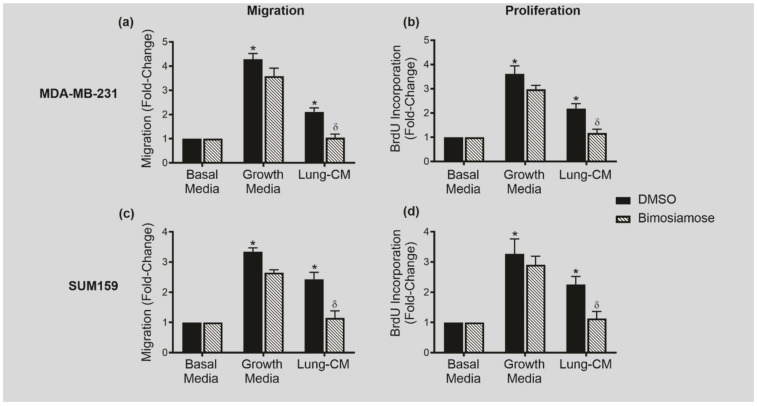
The pan-selectin inhibitor bimosiamose reduces in vitro lung-specific TN breast cancer migration and proliferation. (**a**,**b**) MDA-MB-231 and (**c**,**d**) SUM159 human breast cancer cells were treated with DMSO (vehicle control; black bars) or 1 mM bimosiamose (hatched bars) and subjected to transwell migration (**a**,**c**) or proliferation (**b**,**d**) assays in the presence of basal media, growth media (10% FBS), or lung-CM. Five high-powered fields of view (HP-FOV) were captured per transwell and analyzed using Image J. Data are presented as mean ± SEM; fold-change relative to basal media conditions (*n* = 3). * = significantly different than basal media. δ = significantly different than respective DMSO control.

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
