# Peer review of "Lung-Derived Selectins Enhance Metastatic Behavior of Triple Negative Breast Cancer Cells"

_biomedicines, 2021, doi:10.3390/biomedicines9111580_

Round 1
Reviewer 1 Report
Manuscript was written correctly. In my opinion, it describes the topic presented in detail. As minor point, I suggest to improve the quality of the figures in the manuscript.
Author Response
- “Manuscript was written correctly. In my opinion, it describes the topic presented in detail. As minor point, I suggest improving the quality of the figures in the manuscript”.
We thank the reviewer for this helpful suggestion. We have now made the figures larger and improved their quality in the revised manuscript.
Reviewer 2 Report
The authors presented interesting data on the role of selectins in the metastatic potential of TNBC. However, several points need improvement in this work
You used healthy mice to obtain lung-conditioned media. Have you tested if lung deriving from healthy or cancerous mice contain soluble selectins in different proportion?
Moreover, you attributed the pro-metastatic activity of selectins to the interaction with CD44. Have you tested CD44 expression?
CD44 is expressed by immune cells. Please add a comment about the possible interaction between the role of selectins and immune infiltration of cancer.
In your work you tested only TNBC subtypes. Have you tested also luminal or HER2+ subtypes?
Figure 2. Please add the meaning of Δ or rSELE/P/L, CM, and SEM.
Proliferation of TNBC exposed to Lung-CM and Lung-CM selectin depleted was higher than that of TNBC exposed to basal media. These results seem to be determined by other soluble factors. Which one? Please explain.
You write that in ex vivo experiments you used SUM159 cell line instead of SUM 149, due to the low metastatic potential. Why do you use SUM149 for migration and proliferation assays instead of SUM159 cell line? Considering that the project focuses on metastatic potential it is better to use the SUM 159 cell line for all in vitro experiments.
P.10, Line 380. Please spell DMSO.
Figure 5. Not clear if data with Bimosiamose are significantly different from basal media, considering that the * is only in correspondence of DMSO.
Proliferation and migration are higher with growth media than with lung-CM. This suggests a less important impact of selectins compared to other soluble factors. Please explain.
In Paragraph 3.3 you demonstrated that selectins did not influence breast cancer cells proliferation. In paragraph 3.5 you showed that selectin inhibitions reduce proliferation. How do you explain this discordance?
Experiments with selectin inhibitors in vivo should be done. Please add a comment in discussion section.
The title should focus more on the results shown than on a potential therapeutic role. The therapeutic role still seems to be a long way off as these data are mainly in vitro.
Reviewer 3 Report
Well-done study. One minor correction:
Please add "days" at the end of line 240
Author Response
- “Well-done study. One minor correction: Please add "days" at the end of line 240”.
We thank the reviewer for catching this error and have now corrected it in the revised manuscript (line 244).
Reviewer 4 Report
In this paper, the authors address the question of what drives TNBC metastasis to lung and focused their attention on the potential pro-metastatic role of soluble E-, P- and L-selectins produced by healthy lungs. They evaluated their function in the metastasis process either by studying their capacity to promote TNBC migration to lung-conditioned media in vitro or by analysing the effect of inhibition/knocking-out.
From this point of view, the topic of the paper is clearly of interest since targeting lung microenvironnement is an attractive strategy to design new therapeutic approaches against lung metastasis. The underlying idea is to identify new potential therapeutic targets in the lung microenvironnement and to take advantage of the accessibility of the lungs to inhalable treatments to develop new therapeutic approaches.
However, the paper is a bit frustrating because the authors focus on global results without exploring in more detail the role/importance of each selectine in promoting cancer cell migration to lung-CM, by exploring signaling pathways for instance.
The manuscript is well-written, easy to read and the results are presented appropriately.
Overall, the experimental section is correctly described but some points need to be improved or clarified
- p4 L 174: the secondary antibody is missing in table 2S
- p4 L191-192: In the in vitro cell migration assays, the fixation step is performed before removing the cells that have not migrated from the top of the membrane. Is this correct?
- p4, L187 et figure 1a-c: the concentration units of selectins for rescue conditions are not in agreement (except for E-selectin) with that determined in figure 1a-c: pg/ml versus ng/ml
Major comments
Concerning the direct interaction of mouse selectins with CD44 on human breats cancer cells, it is regrettable that the control input was not on the same immunoblot (Fig. 1.e). Depleting the cell lysate of CD44 could provide further evidence of the selectin/CD44 interaction (disappearance of the 90 kDa band).
In the experiments in figure 2, the fold-change induced by Lung-CM on MDA MB 231 is lower than that on SUM149 which is less metastatic. Is this expected?
Could a cell line that does not express CD44 be a good negative control of migration?
A positive control of migration such as growth medium (basal media +10 or 20% FBS) in fig 2 would have been interesting as it could bring information on the specificity of Lung-CM effect on migration. What happens if selectins are added to the growth medium? does it mimic the effect of lung-CM?
As well, depleting the selectins one by one abrogates the migration down to the negative negative control. To my point of view, this result suggests that each selectin is necessary but not sufficient to promote migration rather than that "individual selectins enhance in vitro migration", as stated by the authors in the conclusion (p10 L407). Can the authors comment this ?
To unravel their individual importance in the migration process, it would have been more appropriate to deplete all of them and then to add them individually or combined. Does this experiment has been done?
Such experiment would also have been interesting as a counterpart to the ex vivo PuMA assay and pan-selectin inhibitor experiments.
Moreover, from figure 1a-c, selectins are not present at the same concentration in the lung-CM, selectin E being more concentrated than the other two. Is this difference in concentration compensated by other phenomena to explain that the depletion of one or the other has the same effect?
Also, the authors conclude from all their experiments that selectins play a role in migration but not on proliferation (p11, L417). However, in figure 5, the pan-selectin inhibitor induced both a decrease in migration and proliferation. Can the authors comment on this?
p11, L433: can the authors explain why they propose a redundant or compensatory function of selectins for each other based on the PuMA and pan-inhibition of selectins experiments? it is far from intuitive
The figures quality should be improved, in particular the typing is very small in figures 1-3.
In figure 1, the input control in fig1e should be on the same immunoblot
The authors should check for figure 4a-b wild-type compared to figure 8 in Targeted Lung-Derived Proteins as a Therapeutic Strategy Against Breast Cancer Metastasis.
Round 2
Reviewer 2 Report
The author have clearly answered to my previous comments. However, the answers to points 6 and 7 do not seem to have been included in the discussion, which, in my opinion, leaves the interpretation of the results incomplete.
Reviewer 4 Report
The authors have made the corrections that were requested and have clarified certain points. However, their answers to more fundamental issues are not totally satisfactory.
For instance, point 10: the fact that SELE depletion produces the same effect as SELP and SELL depletions despite its 50-fold lower concentration would be worth highlighting and discussing.
Their response and the text modifications (lane 432-448) concerning the role of selectins in proliferation (point 11 reviewer 2 and 4) are not fully convincing. Their inhibition or knockout experiments strongly suggest that selectins do influence collectively TNBC proliferation. Elucidating the contribution of each selectin in this process would require, as said by the authors, further experiments. The data from the literature added in the discussion do not really support their hypothesis of a compensation/redundancy of selectins in breast cancers. Would this be specific to breast cancer?
As explained by the authors (reviewer 2 point 6), lung-CM contains many molecules involved in the proliferation and migration of tumor cells. Is it inconceivable that compensatory phenomena to the depletion of one or other of the selectins by these components occur? This aspect is not discussed.
Round 3
Reviewer 4 Report
The changes made to the paper have improved its quality and relevance; results have been more thoroughly detailed and the discussion highlights the issues that remain to be addressed.